# Efficacy and safety of bronchial thermoplasty in clinical practice: a prospective, longitudinal, cohort study using evidence from the UK Severe Asthma Registry

Julie Burn,[1] Andrew J Sims,[1,2] Hannah Patrick,[3] Liam G Heaney,[4] Robert M Niven[5,6]

For numbered affiliations see end of article.

**Correspondence to**
Mrs Julie Burn;
julie.burn@nuth.nhs.uk

## ABSTRACT

**Objectives** Use data from the UK Severe Asthma Registry (UKSAR) to assess the efficacy and safety of bronchial thermoplasty (BT) in routine UK clinical practice and to identify characteristics of 'responders'.

**Design** Prospective, longitudinal, cohort, multicentre registry study.

**Setting** All (11) UK centres performing BT.

**Participants and intervention** Patients receiving BT in the UK between 01/06/2011 and 30/09/2016 who had consented to data entry into UKSAR (n=133). Efficacy data were available for 86 patients with a BT baseline and at least one follow-up record. Safety data were available for 131 patients with at least one BT procedure record.

**Primary and secondary outcome measures** Efficacy: AQLQ, ACQ, EuroQol, HADS anxiety and HADS depression scores, FEV$_1$ (% predicted), rescue steroid courses, unscheduled healthcare visits (A&E/Asthma clinic/GP), hospital admissions and days lost from work/school. Safety: peri-procedural events, device problems and any other safety-related findings. Responder analysis: differences in baseline characteristics of 'responders' (≥0.5 increase in AQLQ at 12 months) and 'non-responders'.

**Results** Following Bonferroni correction for paired comparisons, mean improvement in AQLQ at 12 months follow-up compared with BT baseline was statistically and clinically significant (0.75, n=28, p=0.0003). Median reduction in hospital admissions/year after 24 months follow-up was also significant (−1.0, n=26, p<0.0001). No deterioration in FEV$_1$ was observed. From 28 patients with AQLQ data at BTBL and 12-month follow-up, there was some evidence that lower age may predict AQLQ improvement. 18.9% (70/370) of procedures and 44.5% (57/128) of patients were affected by an adverse event; only a minority were considered serious.

**Conclusions** Improvement in AQLQ is consistent with similar findings from clinical trials. Other efficacy outcomes demonstrated improving trends without reaching statistical significance. Missing follow-up data impacted this study but multiple imputation confirmed observed AQLQ improvement. The safety review suggested BT is being performed safely in the UK.

## Strengths and limitations of this study

► The registry captured detailed morbidity and efficacy outcomes in the largest series of severe asthma patients having bronchial thermoplasty in routine UK clinical care, allowing comparison with outcomes reported from clinical trials.

► Comprehensive coverage of UK clinical practice, with almost 100% UK coverage of BT procedures being carried out post licence, thus avoiding selection bias.

► Improvement in AQLQ at 12 months assessed by paired analysis was confirmed with mixed effects modelling. Results from complete case analysis were confirmed with multiple imputation to account for the effect of missing data.

► Limitations include the lack of a comparator group and changes over time including prescribed drugs, comorbidities, weight and psychological factors which may affect the measured outcomes.

## INTRODUCTION

Bronchial thermoplasty (BT) is an established non-pharmacological treatment for severe asthma. Thermal energy is applied to the airway wall to reduce bronchial smooth muscle, limiting its ability to contract. It is normally delivered in three bronchoscopic procedures, approximately 3–4 weeks apart. Clinical trials,[1–3] and follow-up studies,[4–6] provided early evidence of the efficacy and safety of BT but left uncertainty over whether the trial results would translate into clinical practice.

The need for more evidence from routine practice was identified by a Cochrane Review in 2014,[7] in guidelines jointly published by the European Respiratory Society and the American Thoracic Society,[8] and in the UK by the National Institute for Health and Care Excellence (NICE) in 2012. NICE reviewed BT,[9] and recommended collection of safety

and efficacy outcomes through the UK Severe Asthma Registry (UKSAR).[10]

UKSAR was established in 2006 to collect data on patients with severe asthma and, in response to NICE's recommendations, was expanded to incorporate BT. A recent study of procedural and short-term safety of BT in UK clinical practice using both evidence from UKSAR and routine health data[11] reported that although the rate of adverse events appeared higher than reported from clinical trials, only a minority were considered significant and that the higher rate may be explained by greater severity of asthma in patients being treated in this real-world setting.

Several case series have been published, reporting on BT in clinical practice.[12–17] These and an interim analysis from a large manufacturer-sponsored study[18] all demonstrate some improvement in quality of life, reductions in exacerbations and healthcare resource utilisation and, in general, concur with the findings of the clinical trials that BT is safe, with none raising any serious concerns. However, Thomson and Chanez[19] recently reported on the contributions of the evidence to date in assessing the efficacy of BT in clinical practice and conclude that uncertainty remains.

The main aim of this study is to assess the longer term efficacy for BT in UK clinical practice. Secondary aims are to use experience from clinical practice to identify the characteristics of those patients most likely to benefit from BT, and to update safety evidence reported previously. It uses data collected in UKSAR and is one of the largest 'real-world' observational studies of BT to date, with all UK centres performing BT contributing data.

## METHODS

The study was designed to follow the recommendations of NICE Guidance,[9] and also current international efforts to improve data collection outside of a clinical trial setting for new technologies.[20 21]

### Registry design

Safety and efficacy outcomes and the proposed data fields to be added to the baseline, procedural and follow-up pages in UKSAR were agreed through consultation with clinicians in the UK Severe Asthma Network and with the support of NICE (see online supplementary table S1). Guidance on submitting data to UKSAR was issued to all centres providing BT to achieve consistent data capture. Modifications to UKSAR to include BT therapy were reported in Burn *et al.*[11]

### Inclusion criteria

Eligible patients were those selected to receive BT in the UK between 01/06/2011 and 30/09/2016 with data in UKSAR. Patients included in the efficacy study must have received BT treatment, had a valid BT baseline (BTBL) record and at least one follow-up record.

### Patient and public involvement

Patients and the public were not involved in the design of this study. However, the findings are presented as evidence in the update of NICE interventional guidance on Bronchial Thermoplasty.[9]

### Registry coverage

During the data collection period, regular contact was maintained with the sole manufacturer of the BT device to verify that all (11) UK centres performing BT were entering data into UKSAR. Regular contact with participating centres was maintained to encourage data entry and to confirm procedure numbers.

### Data extraction

Ethics approval for the registry was provided. Patients having BT between June 2011 and September 2016 were invited to give fully informed written consent to record their information on UKSAR. BTBL, procedure and follow-up records in UKSAR at 31 October 2016 were extracted for analysis. In accordance with registry information governance requirements, the data were provided for analysis in anonymised form.

### Data cleaning

The follow-ups nearest in time to 6, 12, 24, 36, 48 and 60 months following the last BT procedure were allocated to these follow-up points and the median times to these were calculated. Count data for rescue steroid courses, unscheduled healthcare visits, hospital admissions and days lost from work or school were annualised to compare rates before and after BT.

### Patient baseline characteristics

Patient baseline characteristics were obtained from UKSAR including: age at first BT procedure, gender, body mass index (BMI), smoking status, eosinophil count (blood), asthma status (including pre-bronchodilator forced expiratory volume in 1s ($FEV_1$) (% predicted) and Asthma Quality of Life Questionnaire (AQLQ) 32 question score[22]), Asthma Control Questionnaire (ACQ) score, EQ-5D-3L descriptive index score, HADS anxiety and depression scores, rescue steroid courses, unscheduled healthcare usage, hospital admissions, days lost from work/school in the previous year, maintenance oral steroids (Y/N), oral steroid dose and anti-IgE medication (Y/N). Characteristics of age, gender, $FEV_1$ and AQLQ score were compared with patients enrolled in clinical trials,[1–3] and characteristics of those included in the efficacy study were compared with those excluded to check for exclusion bias.

### Efficacy outcomes

The efficacy outcomes considered were AQLQ, ACQ, EQ-5D-3L (descriptive index), HADS anxiety and HADS depression scores, $FEV_1$ (% predicted), rescue steroid courses, unscheduled healthcare visits (A&E/Asthma clinic/GP), hospital admissions and days lost from work/school.

Outcome measures at BTBL and follow-up points were calculated using all available data and compared pairwise between BTBL and both 12-month and 24-month follow-ups.

## Potential confounding factors

Types and doses of medications at BTBL and follow-up were not included as outcome measures but were studied as potential confounding factors. For the 86 patients in the efficacy study, the proportions taking maintenance oral steroids and the median doses at BTBL and 12-month follow-up were compared. The numbers of patients who either remained on oral steroids, off oral steroids or started/stopped oral steroids between BTBL and 12-month follow-up were also calculated.

A biologic therapy (Omalizumab) may also have been prescribed as an anti-IgE medication, and the proportions of patients for whom the anti-IgE registry field was ticked at BTBL and 12-month follow-up were compared.

## Statistical analysis

Mean values were calculated for continuous variables and median values for count data. Paired comparisons were performed using: two-sided paired t-tests for continuous variables; two-sided single sample bootstrap hypothesis tests for count data, with data grouped by follow-up time points. All variables were checked for normality using Shapiro-Wilk tests, and p values obtained by t-tests were confirmed using two-sided non-parametric bootstrap tests. Comparison of groups used two sample t-tests for continuous variables and Wilcoxon rank sum test for count data. Comparisons of proportions used proportion test and fisher test. The Bonferroni method was used to adjust for multiple comparisons. Analysis was performed using the 'R' statistical programming language with a significance level of 95%.

## Missing data and additional statistical analyses

To assess the effect of missing data, and potential bias due to using available case analysis, additional statistical analyses were performed.

(1) Multiple imputation was applied to study the change in AQLQ from BTBL to 12-month follow-up. The R package 'MICE'[23] was used to impute missing data values in the records of the 60 patients having both BTBL and 12-month follow-up records. The levels of missing data were examined and the number of imputations used in the model set to 100 (see online supplementary table S2).[24–27]

Paired t-tests were undertaken on each of the 100 imputed data sets from which the mean and SD of the 'mean change in AQLQ' were calculated. The rate of 'mean change in AQLQ' <0.5 (not clinically significant) and the rate of non-significant p value (<0.05) from the 100 imputations were also reported.

(2) A linear mixed effects model was used to measure the change in AQLQ following BT by using all available observations without imputation with the R package 'lmer'.[28] This class of model accounts for repeated observations (multiple follow-ups), tolerates missing values and estimates sizes of fixed and random effects separately. The model response was AQLQ. There was one fixed effect, time, to test whether AQLQ changed over time, and two random effects: of patients on intercept (ie, it accounted for different baseline AQLQ between patients), and of patients on slope (ie. it accounted for different patients having different rates of change of AQLQ after treatment). The null hypothesis of there being no change in AQLQ over time was tested using a parametric bootstrap hypothesis test with a generalised likelihood ratio test statistic. CI were estimated using bootstrapping.

## Comparison of responders with non-responders

Using the AQLQ Minimally Important Clinical Difference (MCID), a responder was defined as ≥0.5 increase in AQLQ score at 12-month follow-up compared with BTBL. The baseline characteristics of responders were compared with non-responders to assess potential characteristics of those most likely to benefit from BT.

A linear model was constructed with change in AQLQ (12-month follow-up – BTBL) as the response variable and four covariates: BT1 age, baseline BMI, baseline $FEV_1$ (% predicted), baseline eosinophil count (blood). Patients with AQLQ observations at BTBL and 12-month follow-up were included. All covariates were centred on their median values.

To assess the effect of missing covariates, the model was repeated using multiple imputation with the records of 60 patients having BTBL and 12-month follow-up records. Median centering was not possible within the multiple imputation model.

## Safety outcomes

UKSAR records added since a previous report of safety outcomes[11] were reviewed, and peri-procedural events, device problems and other potential safety findings were tabulated.

## RESULTS

### Data and study participants

Eighty-six patients who received BT treatment had a valid BTBL record and at least one follow-up record in UKSAR was eligible for inclusion in the efficacy analysis (figure 1). The total follow-up time in person-years from final BT treatment until the data extraction date of 31 October 2016 was 253.73 (median 2.85). Data in UKSAR for 128 patients having at least one BT procedure were available for review of safety outcomes. The total numbers of BT records available in UKSAR are shown in online supplementary table S3.

### Baseline characteristics

One hundred and twenty-six patients had a valid BTBL record (women: 69.8%; mean age at first BT: 43.7±12.2 years (n=125); BMI: 31.8±7.5 kg/m$^2$ (n=119);

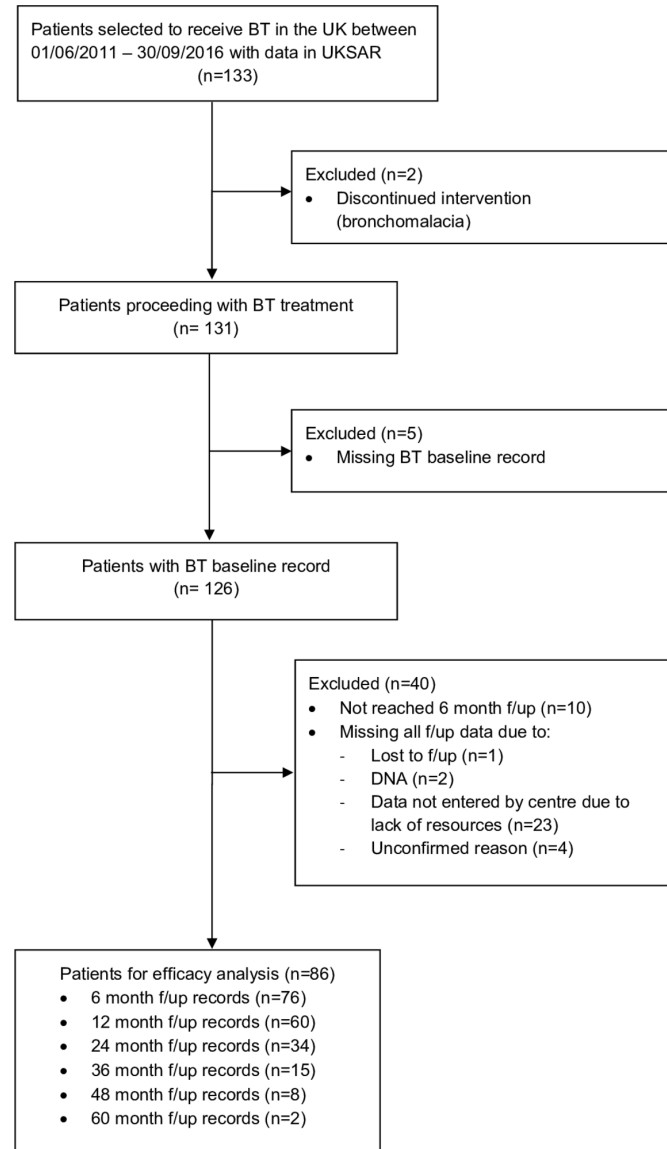

**Figure 1** Efficacy study participants. BT, bronchial thermoplasty; UKSAR, UK Severe Asthma Registry.

non-smoker/ex-smoker: 98.3% (n=117); FEV₁ (% predicted): 71.0±21.8 (n=118)).

The baseline characteristics of patients included in the efficacy study (n=86) and excluded (n=40) are shown in

online supplementary table S4. The mean age of those included was lower than the excluded group (not significant after Bonferroni correction). No other significant differences were found between the two groups.

Compared with published clinical trials,[1–3] patients in this efficacy study were on average older, with lower FEV₁ (% predicted) values (except for the RISA trial) and lower AQLQ scores (table 1).

### Efficacy outcomes

Efficacy outcome measures at BTBL, 12-month and 24-month follow-up points are shown in table 2 (see online supplementary table S5 for data at 6, 36, 48 and 60 months and online supplementary figure S1). There was a clinically significant (>=0.5) improvement in AQLQ at 12 months. There were also improvements in depression scores at 12 months, reduction in healthcare resource utilisations at 12 months and reduction in hospital admissions at 24 months. After Bonferroni correction, only improvement in AQLQ at 12 months and reduction in hospital admissions at 24 months were statistically significant.

Table 3 shows results of paired t-tests following multiple imputation analysis for AQLQ score at 12 months compared with BTBL. 99/100 imputations demonstrated a statistically significant mean increase in AQLQ at 12 months follow-up; 92/100 imputations demonstrated a clinically significant mean increase in AQLQ (>=0.5).

Figure 2 shows the AQLQ repeated measures model for patients with AQLQ score at BTBL and at least one follow-up. From 128 AQLQ observations for 42 patients (median follow-up: 572.4 days), values of the fixed effects were 3.93 (95% CI 3.55 to 4.43) for baseline AQLQ and +0.19 (95% CI 0.00 to 0.36) AQLQ/year for the mean annual change of AQLQ after treatment. The random effect of baseline AQLQ due to individual patients was 1.06 (0.75 to 1.37), the random effect of individual patients on the annual change of AQLQ was 0.15 (0.02 to 0.39) AQLQ/year and the residual error SD in the model was 0.81 (0.67 to 0.92) AQLQ. The null hypothesis of no change in AQLQ over time was rejected at the 5% level (p=0.027). Responders (coloured blue) are those

**Table 1** Baseline characteristics of patients with bronchial thermoplasty (BT) in UK Severe Asthma Registry (UKSAR) compared with clinical trials

|  | UKSAR | AIR2 trial | AIR trial | RISA trial |
|---|---|---|---|---|
| Mean age at BT1 (years) | 43.7±12.2 (n=125) Range 21–74 | 40.7±11.9 (*BT, n=190) 40.6±11.9 (†C, n=98) | 39.4±11.2 (BT, n=55) 41.7±11.4 (C, n=54) | 39.1±13.0 (BT, n=15) 42.1±12.6 (C, n=17) |
| % female | 70 (n=126) | 57 (BT), 61 (C) | 56 (BT), 57 (C) | 60 (BT), 41(C) |
| Pre-bronch FEV₁ (% predicted) | 71.0±21.8 (n=118) Range: 18–109 | 77.8±15.7 (BT) 79.7±15.1 (C) | 72.7±10.4 (BT) 76.1±9.3 (C) | 62.9±12.2 (BT) 66.4±17.8 (C) |
| AQLQ score‡ | 3.66±1.35 (n=81) Range 1.0–6.81 | 4.30±1.17 (BT) 4.32±1.21 (C) | 5.6±1.1 (BT) 5.7±0.9 (C) | 3.96±1.34 (BT) 4.72±1.06 (C) |

*BT, subjects receiving BT treatment.
†C, control subjects.
‡A higher AQLQ score represents better quality of life.
AQLQ, Asthma Quality of Life Questionnaire; FEV₁, forced expiratory volume in 1 s.

**Table 2** Efficacy: all available data in UKSAR at BTBL, 12-month and 24-month follow-up; paired comparisons of BTBL data with outcomes at 12-month and 24-month follow-up (data shown as mean (SD) or median [LQ - UQ])

| | BTBL (n=86) | FU12 (n=60) | FU24 (n=34) | BTBL to FU12 (paired) | BTBL to FU24 (paired) |
|---|---|---|---|---|---|
| AQLQ score | 3.64 (1.26) n=59 | 4.24 (1.45) n=37 | 4.40 (1.62) n=19 | 0.75 n=28 (p=0.0003)* | 0.39 n=16 (p=0.148) |
| EQ-5D score | 0.53 (0.38) n=42 | 0.62 (0.38) n=29 | 0.65 (0.35) n=18 | 0.008 n=18 (p=0.909) | 0.029 n=13 (p=0.706) |
| ACQ score | 3.28 (1.36) n=49 | 2.75 (1.34) n=40 | 3.06 (1.27) n=21 | −0.43 n=36 (p=0.083) | −0.26 n=19 (p=0.370) |
| HADS score (Anxiety) | 8.52 (5.54) n=48 | 6.46 (5.20) n=28 | 5.28 (5.65) n=18 | −1.60 n=20 (p=0.078) | −0.93 n=14 (p=0.216) |
| HADS score (Depression) | 6.46 (5.25) n=48 | 5.07 (4.50) n=28 | 4.67 (4.85) n=18 | −1.60 n=20 (p=0.047) | −0.57 n=14 (p=0.336) |
| FEV1 (% predicted) | 69.65 (21.71) n=82 | 74.90 (21.34) n=52 | 72.71 (21.08) n=31 | 3.51 n=49 (p=0.152) | 2.57 n=30 (p=0.560) |
| Days from last procedure to follow-up | - | 381.5 [363.8–408] n=60 | 738.5 [720.3–775.8] n=34 | - | - |
| Rescue steroid courses (annualised) | 4 [2–5.5] n=75 | 3.1 [1.4–5.8] n=55 | 1.9 [0.8–4.9](n=29) | −0.26 n=49 (p=0.307) | −1.42 n=27 (p=0.255) |
| Unscheduled healthcare† (annualised) | 5 [2–6] n=71 | 3.2 [1.1–5.6] n=52 | 1.3 [0–2.9] (n=29) | −0.93 n=47 (p=0.050) | −1.55 n=24 (p=0.062) |
| Hospital admissions (annualised) | 2 [0–3] n=76 | 0 [0–1.9] n=55 | 0 [0–0.6] n=29 | −2.0 n=51 (p=0.056) | −1.0 n=26 (p<0.0001)* |

*Significant with Bonferroni correction applied for nine paired comparisons (P<0.006) at each follow-up point.
†Includes visits to A&E, GP and asthma clinic.

with an >=0.5 increase in AQLQ at 12-month follow-up compared with BTBL.

There was no significant change in mean values of $FEV_1$ (% predicted) at 12 or 24 months.

The large number of missing data points and zero entries for 'days lost from work/school' prevented reliable analysis. Table 4 shows all available data for days lost from work or school at BTBL (assumed to represent the preceding 12 months) and 12-month follow-up (annualised). Prior to BT, 45.2% of patients (19/42) reported losing at least 1 day per year, compared with 13.5% (5/37) in the year following BT; p=0.005 by proportion test.

## Potential confounding factors

For 86 patients in the efficacy study, the proportion taking maintenance oral steroids at BTBL was 55/85 (64.7%) compared with 36/60 (60%) at 12-month follow-up; p=0.56. The proportion for whom the anti-IgE registry field was ticked (indicating a biologic) at BTBL was 14/84 (16.7%) compared with 9/54 (16.7%) at 12-month follow-up; p=1.0.

Thirty-four patients took maintenance oral steroids at both BTBL and 12-month follow-up, 20 patients did not take them at either BTBL or 12-month follow-up, four patients took them at BTBL but not 12-month

**Table 3** Multiple imputation (MI): paired comparisons of AQLQ at BTBL and 12-month follow-up (FU12) compared with non-imputed data

| | AQLQ score (BTBL) | AQLQ score (FU12) | AQLQ change | P value |
|---|---|---|---|---|
| *Non-imputed data (n=28, mean±SEM) | 3.52±0.21 | 4.27±0.27 | 0.75±0.07 | 0.0003 |
| †MI data (n=60) | 3.58±0.09 | 4.24±0.11 | 0.66±0.13‡ | <0.05§ |

*Paired t-test used to compare 28 patients who had AQLQ measurements at BTBL and FU12.
†60 patients had BTBL and FU12 records which could be used in the multiple imputation. 100 imputations were performed, resulting in 100 values for mean AQLQ score at BTBL and 12 month follow-up, and mean change in AQLQ. The means of these values are presented in the table. Columns included in the imputation: Hospital, Gender, BT1 Age, BL BMI, BL Smoking Status, BL Eosinophil Count (blood), BL FEV1 (% predicted), BL AQLQ, BL Eq_5d, BL ACQ, BL HADS (Anxiety), BL HADS (Depression), BL Rescue Steroid Courses, BL Unscheduled Healthcare Visits, BL Hospital Admissions, FU12 Age, FU12 BMI, FU12 Smoking Status, FU12 Eosinophil Count (blood), FU12 FEV1 (% predicted), FU12 AQLQ, FU12 Eq_5d, FU12 ACQ, FU12 HADS (Anxiety), FU12 HADS (Depression), FU12 Rescue Steroids (annualised), FU12 Unscheduled Healthcare Visits (annualised), FU12 Hospital Admissions (annualised).
‡92/100 imputed datasets had a mean AQLQ change>=0.5.
§99/100 imputed datasets had p<0.05.

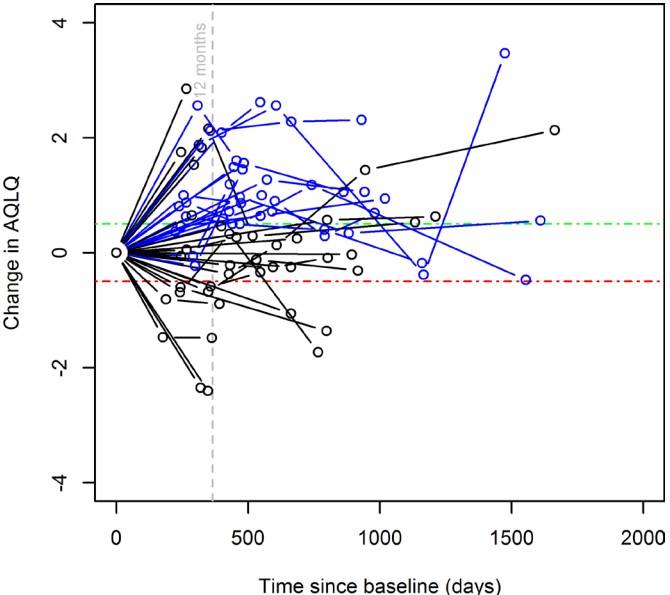

**Figure 2** AQLQ series for patients with AQLQ data at BT baseline and at least one follow-up. Horizontal dashed lines show +/-0.5 (clinically significant) change in AQLQ from BT baseline. Responders (defined as having >=0.5 increase in AQLQ at 12 month follow-up compared with BT baseline) are shown in blue. Non-responders are shown in black.

follow-up, two patients did not take them at BTBL but did at 12-month follow-up. For 31/34 patients taking maintenance oral steroids at BTBL and 12-month follow-up (with valid dose information), the median dose (mg/day) at BTBL was 25 (IQR:15–30) compared with 20 (IQR:10–32.5) at 12-month follow-up; p=0.44.

### Analysis of responders versus non-responders
18/28 patients with AQLQ data at BTBL and 12-month follow-up were classified as responders (defined by

**Table 4** Data in UKSAR for days lost from work/school (self-reported) at BTBL and 12-month follow-up

| Days lost from work/school data | BTBL* | FU12† |
|---|---|---|
| Number of patients with records | 86 | 60 |
| Number of patients with days lost reported | 42 | 37 |
| Patients reporting zero days | 23 | 32 |
| Patients reporting >zero days | 19 (45.2%) | 5 (13.5%) |
| Number of days lost reported | 406 | 61 |
| Median [IQR] days lost/person (for all patients reporting days lost) | 0 [0–10] | 0 [0–0] |
| Median [IQR] days lost/person (for patients reporting>zero days lost) | 10 [7–15] | 13.0 [8.9–15.1] |

*Data at BTBL is assumed to represent a 12 month period.
†Data at FU12 is annualised.

>=0.5 increase in AQLQ); 10/28 patients were classified as non-responders. The baseline characteristics of both groups are shown in table 5.

As reported in the pivotal AIR2 trial,[1] responders in this study also had lower mean baseline AQLQ and higher mean baseline ACQ than non-responders; however, multivariate analysis showed that the only significant predictor of outcome was age at first BT procedure, whereby younger patients were more likely to have a greater improvement in AQLQ at 12-month follow-up (see online supplementary file 2).

Multiple imputation was performed for the 60 patients having records at BTBL and 12-month follow-up. However, multivariate analysis using the pooled imputed data sets just failed to confirm age (p=0.076) as a significant predictor of improvement in AQLQ at 12-month follow-up (see online supplementary file 2).

### Safety outcomes
There were 370 procedure records in UKSAR for 128 patients. Table 6 summarises peri-procedural events, device problems and potential safety-related events found following manual review of all procedure and follow-up records. Overall, 18.9% of procedures and 44.5% of patients were affected by an adverse event.

Some peri-procedural adverse events, although reported in the 'Unanticipated procedural morbidity' field, could be classed as anticipated. Only a minority of procedures were affected by more serious adverse events: inflamed airways/bleeding (1), significant desaturation (1), lung collapse (one slight) (4), left rib fracture (1), metabolic acidosis/lactic acidosis (1), procedure-related bradycardia (1), procedure being stopped/early termination (5).

Nine procedures in eight patients were carried out using general anaesthesia. The median number of activations for all procedures was 34 [min, LQ, UQ, max: 2, 25, 50.75, 115]. For the procedures carried out under general anaesthesia, the median number of activations was 38 [min, LQ, UQ, max: 12, 30, 39, 50].

There were two recorded device problems relating to catheters; one was changed after two activations when a spark was noticed, in another, kinking and infolding were reported after treatment of the right upper lobe.

### DISCUSSION
In this 'real-world' study of BT, we used clinical registry data collected during routine UK clinical practice to analyse efficacy and safety outcomes. All UK centres performing BT were included and patients in this study were not subject to any specific exclusion criteria; their suitability for BT was determined on an individual basis by a multidisciplinary team. Compared with three clinical trials, patients selected to receive BT in clinical practice were, on average, older, had lower baseline $FEV_1$ (except for RISA trial) and lower AQLQ scores.

**Table 5** Responder analysis: baseline characteristics of patients who had AQLQ data at BT baseline and 12-month follow-up (a responder is defined here as having >=0.5 increase in AQLQ at 12-month follow-up compared with BT baseline). Data shown as mean (SD) or median [min, LQ, UQ, max].

| Characteristic | Responders (n=18) | Non-responders (n=10) |
|---|---|---|
| Age at first BT (years) | 35.78 (8.7) n=18 | 50.4 (11.0) n=10 |
| Female (%) | 61.1 n=18 | 60.0 n=10 |
| BMI (kg/m$^2$) | 27.7 (5.2) n=17 | 33.23 (10.7) n=10 |
| Non-smoker/Ex-smoker (%) | 100.0 n=17 | 88.9 n=9 |
| FEV$_1$ (% predicted) | 65.9 (22.1) n=18 | 62.3 (19.1) n=10 |
| Eosinophil count (blood) | 0.29 (0.25) n=13 | 0.51 (0.39) n=8 |
| AQLQ score | 3.48 (1.19) n=18 | 3.59 (1.02) n=10 |
| EQ-5D score | 0.54 (0.42) n=11 | 0.58 (0.39) n=9 |
| ACQ score | 3.59 (1.25) n=14 | 3.07 (0.71) n=9 |
| HADS score (Anxiety) | 6.20 (5.53) n=15 | 10.33 (5.29) n=9 |
| HADS score (Depression) | 4.53 (5.40) n=15 | 7.78 (5.04) n=9 |
| Rescue steroid courses (previous year) | 5 [0,2,6,12] n=17 | 2 [0,1.25,3,7] n=10 |
| Unscheduled healthcare* (previous year) | 4.0 [0,1,6,12] n=16 | 3 [0,2,5,10] n=9 |
| Hospital admissions (previous year) | 1.5 [0,0,3.25,10] n=16 | 0 [0,0,1.5,4] n=10 |
| Days lost from work or school (previous year) | 0 [0,1,5,15] n=9 | 3.5 [0,0,9,15] n=4 |

*Includes visits to A&E, GP and asthma clinic.

The primary outcome for the AIR2 trial was AQLQ at 12 months and it showed an improvement in AQLQ of borderline statistical and below clinical significance compared with sham. The size of the sham placebo response was unprecedented and made interpretation of the findings complex. In our study, for 28 patients having AQLQ data at BTBL and 12-month follow-up, there was a statistically and clinically significant mean improvement in AQLQ at 12-month follow-up compared with BTBL. The reliability of the improvement in AQLQ reported in the AIR2 trial was questioned[29–31]; therefore, we sought further assurance of our results by conducting multiple imputation analysis for 60 patients having both BTBL and 12-month follow-up records. This also showed a statistically and clinically significant mean improvement in AQLQ at 12-month follow-up, consistent with the non-imputed data.

The AQLQ MCID of 0.5 was originally calculated for patients with mild asthma and, in a severe asthma population, may underestimate responders. It is assumed that MCID remains the same over the whole range of the scale; however, patients with lower baseline AQLQ score may perceive an increase of less than 0.5 as meaningful compared with patients starting with a higher score. Using 0.5 as the MCID, we still found a clinically significant mean improvement in AQLQ at 12-month follow-up compared with BTBL.

Use of repeated measures analysis for 42 patients having AQLQ data at BTBL and at least one follow-up addresses, in part, the arbitrary choice of 12 months for conventional paired analysis of change in AQLQ. This approach estimated that AQLQ increased by a mean of +0.19/year (which was significantly greater than zero) after treatment, assuming it increased linearly. The improvement at 1 year predicted by the mean rate (0.19), taking into account all follow-up observations, was less than the observed paired improvement at 12 months (0.75). This suggests that there is still uncertainty over the characteristics of the long-term persistence effect of BT beyond 12 months.

For other efficacy outcomes, the data showed trends for improvement of similar magnitude to the clinical trials; however, the only other statistically significant change after Bonferroni correction was the median reduction in the annual rate of hospital admissions after 24 months. The data indicate a possible reduction in days lost from work/school but we report improvements with caution as the data includes many zero entries.

Although no significant improvement in FEV$_1$ (% predicted) was shown at any follow-up point, a small increase in absolute FEV$_1$ was seen with no subsequent decline, which is consistent with other reported studies.

Median steroid dose before and after BT was not defined as a primary efficacy outcome in our study but is a potential confounder. We found no significant change in median oral steroid dose at 12-month follow-up compared with BTBL for those patients taking oral steroids at both time points; however, caution is advised as patients are prescribed a variety of medications. Likewise, biologics may confound outcomes, but only 16.7% of patients in the efficacy study were noted as being on anti-IgE medication, with the proportion staying constant between BTBL and 12-month follow-up.

**Table 6** Safety

| Peri-procedural events | Number of procedures affected (n=370*) | Number of patients affected (n=128†) |
|---|---|---|
| Events preventing treatment completion (Excessive cough, discomfort and pain/bronchospasm) | 5 | 5 |
| Infection | 8 | 8 |
| Exacerbation | 13 | 11 |
| Asthma-related symptoms (Drop in FEV1, wheeze, shortness of breath, low Sao2) | 24 | 19 |
| Procedure related symptoms (Bronchospasm, dry cough, chest twinges/tightness/discomfort/pain) | 20 | 16 |
| Other (Left rib fracture) | 1 | 1 |
| Other (Metabolic acidosis) | 1 | 1 |
| Other (Inflamed airways, bleeding medial basal) | 1 | 1 |
| Other (Lung collapse, one slight) | 4 | 4 |
| Other (Procedure-related bradycardia) | 1 | 1 |
| **Additional reports** | | |
| Device-related (catheters needed replacement) | 2 | 2 |
| Prolonged stay (>=7 days) with no reason given) | 3 | 2 |
| A&E attendance with no details given | 1 | 1 |
| Airway tracheomalacia reported | 2 | 2 |
| BT3 postponed 2 months due to inflamed airways & pain | 1 | 1 |
| **Three procedures not able to be performed: ‡** | | |
| BT1 only | - | 2 |
| BT1 and BT2 only | - | 2 |
| **CT scan reports (up to 6 month follow-up) (From 24 CT scan reports in 21 patients)** | | |
| Central bronchiectasis | - | 1 |
| Other bronchiectasis | - | 2 |
| **Total events** | 87 | - |
| **Unique procedures/patients affected** | 70 (18.9%) | 57 (44.5%) |

*370 procedures in total were entered with a date in UKSAR, but some had minimal procedural information recorded.
†Three patients were excluded as they had no procedure records entered in UKSAR.
‡In addition, four patients hadn't completed all 3 procedures at 30/09/2016.

An exploratory responder analysis with four covariates (no interactions) was used to identify patients who might potentially benefit from BT, comparing 18 responders with 10 non-responders. This showed interesting trends, but only younger age was a significant predictor of improvement in AQLQ. Multiple imputation failed to reproduce this finding.

Peri-procedural and short-term safety outcomes were previously reported using data from UKSAR and HES.[11] It was found that 20.4% of procedures in the cohort were affected by at least one reported event (including procedural complication, emergency respiratory readmission or respiratory A&E attendance (without subsequent hospital admission) within 30 days).

This study sought to provide an update using the data available in UKSAR. We found that 18.9% of procedures and 44.5% of patients were affected by at least one event that could be identified from procedure or follow-up records. These rates were calculated using all 370 procedure records with at least the date field completed. However, some of these had no further details of the procedure so their effect on the rates is unknown. Conversely, many peri-procedural adverse events, although reported in the 'Unanticipated procedural morbidity' field, could have been classed as anticipated. Including only the more serious or genuinely unanticipated events in the safety analysis would decrease the rate of adverse events. Hence, full details of all events were included to enable readers to judge the number and severity of events.

A strength of this study is the comprehensive coverage reported from clinical practice. We achieved almost 100% UK coverage of BT procedures being carried out post licence as confirmed by the sole manufacturer of the BT device. Data were obtained from centres performing lower numbers of procedures and those that had previously participated in the clinical trials, thus avoiding any selection bias.

We used more than one statistical method to analyse the change in AQLQ, and the results were consistent in each.

A limitation of this study is the lack of comparator group so the observed improvements may be due to placebo effect. Additionally, over time, various factors affect the measured outcomes; patients are prescribed different drugs after BT, and comorbidities, weight change, and psychological factors may all affect outcomes. We did not study paired data beyond 24-month follow-up as there were diminishing numbers of follow-up records available for analysis and other confounding factors may become more significant.

Despite the recommendation in NICE Guidance for data collection, the biggest limitation is the poor follow-up data completion, resulting in only a small number of patients with data sets available for follow-up analysis. While we recognise that missing data limit generalisability, we performed multiple imputation analysis to address this and the imputation model did support the observed improvement in AQLQ score at 12-month follow-up. Improvements in several other outcome measures which were non-significant after correcting for multiple testing suggest that there is still a need to capture more longer-term outcome information to fully understand the effectiveness of BT.

## CONCLUSIONS

In summary, this paper presents outcomes from the largest UK 'real-world' study of bronchial thermoplasty to date. The mean improvement in AQLQ at 12 months compared with BTBL is consistent with similar findings from clinical trials. No deterioration in $FEV_1$ (% predicted) was observed following BT and there was a significant reduction in hospital admissions at 24-month follow-up. However, improvements were not seen in all patients and an exploration of the characteristics of 'responders' to BT could only identify age as a possible predictor of outcome. Current and future studies looking into factors which determine those who respond well to BT will be crucial to informing patient selection. Continued data collection would also help to understand whether the improvement in AQLQ observed for some patients is maintained over a longer period. Although a health economic analysis was beyond the scope of this study, the improvement in QoL and reduction in healthcare usage would suggest that BT may bring cost savings.

Of the adverse events recorded, only a minority were considered significant by the treating clinician. It appears that BT is being carried out safely in the UK; however, long-term safety should continue to be monitored and the decision to select a patient for BT should continue to lie with a multidisciplinary team who can assess their history and suitability for the procedure.

**Author affiliations**
[5]Division of Infection, Immunity & Respiratory Medicine, Manchester Academic Health Science Centre, Manchester, UK

[1]Northern Medical Physics and Clinical Engineering, The Newcastle upon Tyne Hospitals NHS Foundation Trust, Newcastle upon Tyne, UK
[2]Institute of Cellular Medicine, University of Newcastle upon Tyne, Newcastle upon Tyne, UK
[3]Observational Data Unit, National Institute for Health and Care Excellence, London, UK
[4]Centre for Experimental Medicine, Queens University Belfast, Belfast, UK
[6]North West Lung Centre, Wythenshawe Hospital, Manchester University NHS Foundation Trust, Manchester, UK

**Acknowledgements** We are grateful for the support of staff at centres performing bronchial thermoplasty and completing data entry to the UK Severe Asthma Registry: Royal Brompton Hospital (London), Manchester University NHS Foundation Trust, Gartnavel General Hospital (Glasgow), Freeman Hospital (Newcastle upon Tyne), Heartlands Hospital (Birmingham), University Hospitals of Leicester, University Hospital Southampton, Aintree University Hospital (Liverpool), Queen Alexandra Hospital (Portsmouth), Addenbrooke's Hospital (Cambridge), Nottingham City Hospital. We are also grateful for the support of Sally Welham (British Thoracic Society) who provided invaluable advice to the steering group.

**Contributors** LGH is founder and coordinator of the UK Severe Asthma Registry Data and provided data extracts. JB, AJS, HP, LGH and RMN contributed equally to the design of the study. JB and AJS were responsible for analysis of the data. JB, AJS, HP, LGH and RMN were responsible for interpretation of the analysis. JB, AJS, HP, LGH and RMN drafted, revised and approved the final version of the manuscript.

**Funding** Pilot funding for the UK Severe Asthma Registry was provided as unrestricted research grants from Astra Zeneca, GlaxoSmithKline, Novartis and Medimmune. The extension of the UK Severe Asthma Registry to include bronchial thermoplasty data collection was funded by NICE. JB and AJS are employed by The Newcastle upon Tyne Hospitals NHS Foundation Trust which hosts an External Assessment Centre funded by NICE.

**Competing interests** LGH has received grant funding from Medimmune, Novartis UK, Roche/Genentech Inc, Astra Zeneca and Glaxo Smith Kline, have taken part in Advisory Boards and given lectures at meetings supported by Glaxo Smith Kline, Respivert, Merck Sharpe & Dohme, Nycomed, Boehringer Ingelheim, Novartis and Astra Zeneca. LGH has received support funding to attend International Respiratory meetings (Astra Zeneca, Chiesi, Novartis, Boehringer Ingelheim and Glaxo Smith Kline) and has taken part in asthma clinical trials (GSK, Schering Plough, Synairgen and Roche/Genentech) for which his Institution was remunerated. LGH is Academic Lead for the Medical Research Council Stratified Medicine UK Consortium in Severe Asthma which involves Industrial Partnerships with Amgen, Johnson & Johnson, Genentech / Roche, Astra Zeneca / Medimmune, Aerocrine and Vitalograph. RN was PI on several of the thermoplasty trials, and has received honoraria for lecturing & attending advisory boards from Boston Scientific.

**Patient consent for publication** Not required.

**Ethics approval** Ethics approval for the UK Severe Asthma Registry (UKSAR) was provided by the Office for Research Ethics Committees Northern Ireland (10/NIR02/37).

**Provenance and peer review** Not commissioned; externally peer reviewed.

**Data sharing statement** The registry data used in this study were provided to The Newcastle upon Tyne Hospitals NHS Foundation Trust following approval by the Registry steering committee. In accordance with registry information governance requirements, the data were provided for analysis in anonymised form. The authors are not permitted to make the data publicly available as there is a risk that patients could be identified due to the small numbers involved.

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
