## [Reviewer comments · BMJ Open]

This paper was submitted to a another journal from BMJ but declined for publication following peer review. The authors addressed the reviewers' comments and submitted the revised paper to BMJ Open. The paper was subsequently accepted for publication at BMJ Open.

ARTICLE DETAILS

TITLE (PROVISIONAL)	Efficacy and safety of bronchial thermoplasty in clinical practice: a prospective, longitudinal, cohort study using evidence from the UK Severe Asthma Registry
AUTHORS	Burn, Julie; Sims, Andrew; Patrick, Hannah; Heaney, Liam; Niven, Robert

VERSION 1 - REVIEW

REVIEWER	Dr Robin Gore Cambridge University Hospitals NHS Foundation Trust, UK Received non-monetary support for career development seminars from Novartis UK 2017.
REVIEW RETURNED	17-Oct-2018

GENERAL COMMENTS	This paper is useful, since it supplies some answers which are currently not adequately addressed in the literature: The outcomes from bronchial thermoplasty in patients with SEVERE asthma. Overall, the description of the methods, results and conclusions are accurate and balanced. The conclusions in particular are well-drawn. This manuscript will be useful for physicians treating patients with severe asthma, and will help them to explain the current state of knowledge of patient selection and benefits to patients. In particular, this work helps put the AIR-2 trial into context, since AIR-2 patients were milder than the patients treated in this UK study. A particular strength is the pan-UK selection. I raise the following points: Was there any data on median steroid dose at BTBL and follow-up? Table S4 - for continuous data are these means and SD? - Please clarify.
---

REVIEWER	Jose Cardenas-Garcia University of Michigan USA
REVIEW RETURNED	24-Oct-2018

GENERAL COMMENTS	Nicely written paper, I have some questions for the authors:  - In Table 6, it is mentioned that 4 patients had incomplete BT sessions (370 procedures in 128 patients) , and they should be excluded, thus statistical analysis might need to be performed again as might impact conclusions. If authors disagree, please explain. - Correct Table 6 row 4. - Any records of how BT was performed? General anesthesia or conscious sedation. This might impact the number of activations (and indirectly the efficacy) ? Please discuss this.
--

REVIEWER	Matthew Masoli Royal Devon and Exeter Hospital, UK
REVIEW RETURNED	07-Dec-2018

GENERAL COMMENTS	The study sets out to describe the clinical experience of bronchial thermoplasty within UK severe asthma centres using the UK severe asthma registry. The manuscript is well written, clearly set out and straight forward to read. The data collected by UKSAR is comprehensive for a registry and the discussion is balanced with mention made of the obvious limitations associated with using registry data. The advantage of this study is that this is real life data and the UKSAR has a track record of collecting robust data over many years. There are obvious limitations which have been identified in the paper such as the lack of comparator group and the fact that improvements in outcomes may relate to other aspects of specialist care the patients receive within a UK severe asthma service. However, there is a sustained improvement in AQLQ and HCU. I would be interested in comparisons with QoL data using patients on biologics which the UKSAR is no doubt also collecting. Current biologic trials in severe asthma have struggled to demonstrate clinically significant improvements in QoL. Although most patients gain benefit in QoL with BT I note some patients have a significant worsening in AQLQ and there is still some uncertainty as to which phenotype of severe asthma is more likely to benefit from this treatment. This study provides no clarity on that issue for clinicians. The paper highlights the current UK experience with bronchial thermoplasty. Given the comprehensive nature of the UK severe asthma registry I would agree that this gives the best current data with which to review BT. There are two aspects to the study.  1. efficacy N=86 2. safety N=128 It is pleasing that health related quality of life has been given prominence as a primary outcome measure as it is often neglected by pharma. I think this fits well with a patient centred approach to asthma management which is gaining prominence particularly in European circles. The safety data is useful and well presented. As with any new novel treatment there is some caution but I think with BT this has been more so given it is a mechanical intervention (one off) rather than a
---

	drug that can be stopped at any point. I think readers will find this paper helpful and reassuring. Major points:  1. The patient demographics does not state how many patients were on a biologic. Were any patients started on a biologic agent post BT as this would obviously potentially affect outcomes? 2. Were any patients on maintenance oral steroids and if so what was the OCS reduction? Minor points:  1. The quality of life measures used (disease specific = AQLQ, generic = Euroqol could be slightly more detailed. Is it the mini AQLQ or the 32 question AQLQ? This has a bearing on MCID accuracy. Is it the EQ5D 5L that has been used (was it just the descriptive or was there VAS data)? When considering the responder analysis it may be useful to perhaps discuss some of the potential limitations. Given that the AQLQ was not designed specifically for severe asthma and the calculation of the MCID was made using the 32 question AQLQ in patients with mild asthma (many of whom were not even on an ICS). The MCID was then extrapolated to the mini-AQLQ. Therefore, it's worth bearing in mind that the MCID may not be completely accurate in this patient population and with the mini-AQLQ (i.e. it's more likely to underestimate responders as the bar may well be too high for what patients perceive as a clinically meaningful response). The other issue with historical MCID calculations is whether the patients' perceptions of a clinically meaningful response change with severity of disease. The issue of transitivity – i.e., the assumption that an MCID remains the same over the whole range of the scale. E.g. If a patient already has a good QoL will they value improvement to the same degree as a patient with a poor QoL. This may be more relevant given that the baseline AQLQ in the BT trials ranges from 3.9 to 5.6/7 and 3.6 in UKSAR. I raise it as a point worth mentioning in the paper if one is going to use a HRQoL measure effectively particularly as a primary outcome measure and undertake responder analyses based on the MCID. It may be worth just stating that the AQLQ may not be as sensitive in severe asthma and therefore may underestimate effect.  2. Is there a possible health economic analysis given the QoL benefit and HCU reduction? The benefits appear to be sustained to 2 years although the lack of complete follow up data is a limitation. Overall I think this paper provides a good contribution to the literature.
--	--

REVIEWER	Rita Amaral Faculdade de Medicina da Universidade do Porto, Portugal
REVIEW RETURNED	11-Feb-2019

GENERAL COMMENTS	This study aimed to assess the long-term efficacy for Bronchial Thermoplasty in UK clinical practice, providing a perspective from clinical practice to identify the characteristics of patients who could most benefit from BT.
--

	It is an important topic and the study provide a comprehensive coverage of UK clinical practice and BT procedures. However, there are some methodologic issues that should be clarify and improved.  1. The introduction is well written but last paragraph needs to be a bit shorter. Specifically, the last sentence appears to belong to the methods section rather the paragraph where the study aims/hypotheses are described. 2. Inclusion criteria and data preprocessing are well described. However, in statistical analysis the authors only used parametric tests for continuous variables. Did the authors check the data distribution? If the normality of the distribution is not assumed, then non-parametric test must be applied, and results must be presented accordingly. 3. To facilitate the reader, I suggest the presentation of a graph with the long-term changes of the main continuous variables (e.g. error bar with 95% confidence intervals. 4. Using only measures of central tendency is not the most adequate to describe data. Please add the respective measures of dispersion. 5. Despite the authors described the use of cut-offs, such as the ≥ 0.5 increase in AQLQ at month follow-up, it is not clear that this is considered minimal clinically important difference. Did the authors considered the concept of the MCID? If yes, please mention in the manuscript and discuss accordingly. 6. I understand that the selection bias was minimized, however, the authors must discuss the generalization of the findings or present a power calculation.
--	--

VERSION 1 – AUTHOR RESPONSE

Reviewer: 1

Reviewer Name: Dr Robin Gore

Institution and Country: Cambridge University Hospitals NHS Foundation Trust, UK

Please state any competing interests or state 'None declared': Received non-monetary support for career development seminars from Novartis UK 2017.

This paper is useful, since it supplies some answers which are currently not adequately addressed in the literature: The outcomes from bronchial thermoplasty in patients with SEVERE asthma.

Overall, the description of the methods, results and conclusions are accurate and balanced. The conclusions in particular are well-drawn.

This manuscript will be useful for physicians treating patients with severe asthma, and will help them to explain the current state of knowledge of patient selection and benefits to patients.

In particular, this work helps put the AIR-2 trial into context, since AIR-2 patients were milder than the patients treated in this UK study.

A particular strength is the pan-UK selection.

Dear Dr Gore,

Thank you for your review and positive comments.

I raise the following points:

Was there any data on median steroid dose at BTBL and follow-up?

Measuring steroid dose before and after BT was not one of our prospective outcome measures, however we acknowledge that this may be a confounding factor and have therefore added a section for 'Potential confounding factors' into the methods and results. We have included a comparison of the proportions of patients who were taking maintenance oral steroids at BTBL and 12 month follow-up as well as comparing the median steroid dose for those patients who were taking oral steroids at both time points. We have also included the numbers of patients who either remained on oral steroids, off oral steroids or started/stopped oral steroids between BTBL and 12 month follow-up.

Table S4 - for continuous data are these means and SD? - Please clarify.
Thank you for pointing out this omission. The title caption has been modified.
End of review.

Reviewer: 2

Reviewer Name: Jose Cardenas-Garcia

Institution and Country: University of Michigan, USA

Please state any competing interests or state 'None declared': None declared

Nicely written paper, I have some questions for the authors:

Dear Dr Cardenas-Garcia,

Thank you for your review and positive comments.

- In Table 6, it is mentioned that 4 patients had incomplete BT sessions (370 procedures in 128 patients), and they should be excluded, thus statistical analysis might need to be performed again as it might impact conclusions. If authors disagree, please explain.

The 4 patients who had incomplete BT sessions were included as we followed ITT analysis rather than per protocol for the study. They also fulfilled the inclusion criteria, having received BT treatment, had a valid BT baseline (BTBL) record and at least one follow-up record. ITT analysis is preferable in this study as it avoids overoptimistic estimates of the efficacy of the treatment.

- Correct Table 6 row 4.

Thank you for pointing this out. We have modified 'sob' to 'shortness of breath'

- Any records of how BT was performed? General anaesthesia or conscious sedation. This might impact the number of activations (and indirectly the efficacy)? Please discuss this.

We acknowledge that the use of general anaesthesia is relevant and could potentially impact the number of activations. There were 9 procedures in 8 patients reported to have used general anaesthesia. The median number of activations for all procedures was 34 (min 2 – max 115). For the procedures carried out under general anaesthesia the median number of activations was 38 (min 12 – max 50). We have added this information into the 'Safety outcomes' section.

Reviewer: 3

Reviewer Name: Matthew Masoli

Institution and Country: Royal Devon and Exeter Hospital, UK

Please state any competing interests or state 'None declared': none

The study sets out to describe the clinical experience of bronchial thermoplasty within UK severe

asthma centres using the UK severe asthma registry.

The manuscript is well written, clearly set out and straight forward to read. The data collected by UKSAR is comprehensive for a registry and the discussion is balanced with mention made of the obvious limitations associated with using registry data.

The advantage of this study is that this is real life data and the UKSAR has a track record of collecting robust data over many years. There are obvious limitations which have been identified in the paper such as the lack of comparator group and the fact that improvements in outcomes may relate to other aspects of specialist care the patients receive within a UK severe asthma service. However, there is a sustained improvement in AQLQ and HCU. I would be interested in comparisons with QoL data using patients on biologics which the UKSAR is no doubt also collecting. Current biologic trials in severe asthma have struggled to demonstrate clinically significant improvements in QoL.

Although most patients gain benefit in QoL with BT I note some patients have a significant worsening in AQLQ and there is still some uncertainty as to which phenotype of severe asthma is more likely to benefit from this treatment. This study provides no clarity on that issue for clinicians.

The paper highlights the current UK experience with bronchial thermoplasty. Given the comprehensive nature of the UK severe asthma registry I would agree that this gives the best current data with which to review BT.

There are two aspects to the study.

1. efficacy N=86
2. safety N=128

It is pleasing that health related quality of life has been given prominence as a primary outcome measure as it is often neglected by pharma. I think this fits well with a patient centred approach to asthma management which is gaining prominence particularly in European circles.

The safety data is useful and well presented. As with any new novel treatment there is some caution but I think with BT this has been more so given it is a mechanical intervention (one off) rather than a drug that can be stopped at any point. I think readers will find this paper helpful and reassuring.

Dear Dr Masoli,

Thank you for your review and positive comments.

Major points:

1. The patient demographics does not state how many patients were on a biologic. Were any patients started on a biologic agent post BT as this would obviously potentially affect outcomes?

We acknowledge that this is a potential confounding factor and have added details into a new 'Potential confounding factors' section in the methods and results. There is a registry (Yes/No) field for anti-IgE medication, so we have included a comparison of the proportions of patients for whom this was ticked at BTBL and 12 month follow-up.

2. Were any patients on maintenance oral steroids and if so what was the OCS reduction?

Some patients were on maintenance oral steroids, but measuring steroid dose before and after BT was not one of our pre-set objectives. However we acknowledge that this is also a potential confounding factor and have included it in the new 'Potential confounding factors' section in the methods and results. We have included a comparison of the proportions of patients who were taking maintenance oral steroids at BTBL and 12 month follow-up as well as comparing the median steroid dose for those patients who were taking oral steroids at both time points. We have also included the numbers of patients who either remained on oral steroids, off oral steroids or started/stopped oral steroids between BTBL and 12 month follow-up.

Minor points:

1. The quality of life measures used (disease specific = AQLQ, generic = Euroqol could be slightly more detailed. Is it the mini AQLQ or the 32 question AQLQ? This has a bearing on MCID accuracy.

The 32 question AQLQ was used. We have added this detail to the 'Methods' section.

Is it the EQ5D 5L that has been used (was it just the descriptive or was there VAS data)?

The EQ-5D-3L was used. This work was commissioned by NICE, and as the EQ-5D VAS is not generally accepted by decision-making bodies including NICE we did not include it in our study. Only the descriptive index score is included in the analysis.

When considering the responder analysis it may be useful to perhaps discuss some of the potential limitations. Given that the AQLQ was not designed specifically for severe asthma and the calculation of the MCID was made using the 32 question AQLQ in patients with mild asthma (many of whom were not even on an ICS). The MCID was then extrapolated to the mini-AQLQ.

Therefore, it's worth bearing in mind that the MCID may not be completely accurate in this patient population and with the mini-AQLQ (i.e. it's more likely to underestimate responders as the bar may well be too high for what patients perceive as a clinically meaningful response).

The other issue with historical MCID calculations is whether the patients' perceptions of a clinically meaningful response change with severity of disease. The issue of transitivity – i.e., the assumption that an MCID remains the same over the whole range of the scale. E.g. If a patient already has a good QoL will they value improvement to the same degree as a patient with a poor QoL. This may be more relevant given that the baseline AQLQ in the BT trials ranges from 3.9 to 5.6/7 and 3.6 in UKSAR.

I raise it as a point worth mentioning in the paper if one is going to use a HRQoL measure effectively particularly as a primary outcome measure and undertake responder analyses based on the MCID. It may be worth just stating that the AQLQ may not be as sensitive in severe asthma and therefore may underestimate effect.

We agree that this is a very relevant point and we have added a paragraph into the discussion.

2. Is there a possible health economic analysis given the QoL benefit and HCU reduction? The benefits appear to be sustained to 2 years although the lack of complete follow up data is a limitation.

Yes there is scope for a health economic analysis although that is beyond the scope of this study. We have added a sentence to this effect in the conclusion.

Overall I think this paper provides a good contribution to the literature.

Thank you again for your positive review.

Reviewer: 4

Reviewer Name: Rita Amaral

Institution and Country: Faculdade de Medicina da Universidade do Porto, Portugal

Please state any competing interests or state 'None declared': None declared

This study aimed to assess the long-term efficacy for Bronchial Thermoplasty in UK clinical practice, providing a perspective from clinical practice to identify the characteristics of patients who could most benefit from BT.

It is an important topic and the study provide a comprehensive coverage of UK clinical practice and BT procedures. However, there are some methodologic issues that should be clarify and improved.

Dear Dr Amaral,

Thank you for your review and positive comments.

1. The introduction is well written but last paragraph needs to be a bit shorter. Specifically, the last sentence appears to belong to the methods section rather the paragraph where the study aims/hypotheses are described.

Thank you for your suggestion. We have moved the last sentence into the Methods section.

2. Inclusion criteria and data preprocessing are well described. However, in statistical analysis the authors only used parametric tests for continuous variables. Did the authors check the data distribution? If the normality of the distribution is not assumed, then non-parametric test must be applied, and results must be presented accordingly.

We checked for normality of continuous data using Shapiro-Wilk tests. In two of the 10 time-point comparisons there was weak evidence for non-normality, and we repeated the paired testing (for all comparisons) using non-parametric bootstrap tests and confirmed that the reported P-values were robust. This has been clarified in the methods section.

3. To facilitate the reader, I suggest the presentation of a graph with the long-term changes of the main continuous variables (e.g. error bar with 95% confidence intervals).

We have produced a graph and this is included in the Supplementary file as Fig S1

4. Using only measures of central tendency is not the most adequate to describe data. Please add the respective measures of dispersion.

We have added the upper and lower quartile information to results tables in the main document and values for min, lower quartile, upper quartile and max in the Supplementary file.

5. Despite the authors described the use of cut-offs, such as the ≥ 0.5 increase in AQLQ at month follow-up, it is not clear that this is considered minimal clinically important difference. Did the authors considered the concept of the MCID? If yes, please mention in the manuscript and discuss accordingly.

We agree that this is a relevant point and we have added reference to the MCID to the methods section and the discussion.

6. I understand that the selection bias was minimized, however, the authors must discuss the generalization of the findings or present a power calculation.

We considered that a power calculation was not appropriate in this study as we included all the eligible patients – we didn't set out to achieve an effect size. We have added a sentence into the discussion to explain that whilst we recognise that missing data limits generalisability, we performed multiple imputation analysis to address this.

VERSION 2 – REVIEW

REVIEWER	Jose Cardenas-Garcia University of Michigan - Ann Arbor
REVIEW RETURNED	12-Apr-2019

GENERAL COMMENTS	Appreciate and accept the answers of my previously raised points. My recommendation is accept to publish.
--

REVIEWER	Rita Amaral Faculty of Medicine of University of Porto, Portugal
REVIEW RETURNED	27-Mar-2019

GENERAL COMMENTS	This comprehensive revision is responsive to reviewer comments on the initial submission and is substantially improved. I have no further comments or concerns.
---